# Hepatitis C virus cascade of care among adults in Sindh province, Pakistan: Findings from 2019–2020 household sero-survey

Tesfa Sewunet Alamneh[1,2]*, Josephine G. Walker[1], Aaron G. Lim[1], Ejaz Alam[3], Saeed Hamid[4], Graham R. Foster[5], Naheed Choudhry[5], M. Azim Ansari[6], Huma Qureshi[7], Peter Vickerman[1]

**1** Population Health Sciences, Bristol Medical School, University of Bristol, Bristol, United Kingdom, **2** Department of Epidemiology and Biostatistics, Institute of Public Health, College of Medicine and Health Sciences, University of Gondar, Gondar, Ethiopia, **3** Pakistan Health Research Council Research Centre, Karachi, Pakistan, **4** Aga Khan University, Karachi, Pakistan, **5** Queen Mary University of London, London, United Kingdom, **6** Nuffield Department of Medicine, University of Oxford, Oxford, United Kingdom, **7** Consultant Gastroenterologist, Doctor Plaza, Clifton Karachi, Pakistan

\* tesfa.alamneh@bristol.ac.uk

## Abstract

Pakistan has the largest national burden of hepatitis C virus (HCV) infections (9.8 million). High levels of testing and treatment are needed to achieve HCV elimination, but little data exists on this in Pakistan. A household sero-survey from Sindh province (2019–2020) collected self-reported data from adults on previous HCV testing and treatment, and undertook HCV-antibody (HCV-Ab) testing of participants (2988 children (<18) and 3684 adults) and HCV-RNA testing of HCV-Ab positive individuals. We determined the self-reported HCV cascade-of-care among adults ever eligible for HCV treatment, defined as either having a past infection (HCV-Ab positive and HCV-RNA negative) with self-reported treatment history or current infection (HCV-RNA positive). We assessed factors associated with self-reporting ever being HCV-tested using multi-variable logistic regression. Overall, 10.8% (397/3684) of adults tested HCV Ab-positive in the sero-survey, of which 80.9% (321/397) had a HCV-RNA test result. Of adults defined as ever treatment eligible (n=232), 40.9% (95/232) reported a previous HCV test and 91.2% (87/95) reported testing positive. Of these, HCV treatment was reported by 69.0% (60/87) and 46.7% (28/60) of treated individuals tested HCV-RNA-negative. Overall, 25.9% (60/232) of treatment-eligible adults reported being treated. The regression analysis suggested that males, older adults (>25 years), and adults with a secondary or higher education level were more likely to have ever been tested for HCV, as were individuals with a family history of hepatitis, received HBV vaccination or that had various risk factors linked to HCV transmission (e.g., blood transfusion, having tattoo/acupuncture, hospitalisation or therapeutic injection (s) history). The cascade-of-care for HCV needs improving to eliminate HCV in Pakistan, especially among younger adults, women and people with low education levels.

**Data availability statement:** All relevant data are included in the manuscript and supporting information. The STATA analysis code is publicly available at the following repository: https://github.com/Tesfa-Alamneh/HCV-CoC-analysis-code-.git.

**Funding:** This work was supported by the NIHR Health Protection Research Unit (HPRU) in Behavioural Science and Evaluation at the University of Bristol (NIHR200877 to AGL and PV); the Wellcome Trust (WT220866/Z/20/Z to JW, AGL, and PV); and the Sir Henry Dale Fellowship jointly funded by the Royal Society and the Wellcome Trust (220171/Z/20/Z to MAA). Additional support was provided by the NIHR (NIHR133208 to JW, AGL, and PV). TSA acknowledges the PhD scholarship from the DESTINE project funded by the NIHR (NIHR133208) using UK aid from the UK Government to support global health research. The views expressed in this publication are those of the author(s) and not necessarily those of the NIHR or the UK Department of Health and Social Care. The funders had no role in study design, data collection and analysis, decision to publish, or preparation of the manuscript.

**Competing interests:** The authors have declared that no competing interests exist.

**Abbreviations:** HCV, hepatitis C virus; CoC, Cascade of Care; HCV-Ab, HCV antibody; aOR, adjusted odds ratios: 95%CI, 95% confidence intervals;IDU, injectable drug use;DAAs, direct-acting antivirals; WHO, World Health Organisation; GHSS, Global Health Sector Strategy; LMICs, low- and middle-income countries; NAT, nucleic acid testing; HBV, hepatitis B virus; SVR, sustained virologic response.

## Background

Hepatitis C virus (HCV) infection is a major cause of chronic liver disease and associated morbidity [1]. Globally, approximately 58 million people were chronically infected with HCV in 2019, with 1.5 million new infections and 290,000 deaths occurring annually [2]. HCV is transmitted through contaminated blood exposure, with the main modes of transmission being through unsafe healthcare practices and injection drug use (IDU) [3–7]. The distribution of HCV infection varies considerably across countries [3]. Pakistan, with an estimated 9.8 million people living with chronic HCV infection, may now have the largest burden of infection globally following Egypt dramatically reducing their burden of infection [8].

The emergence of highly effective direct-acting antivirals (DAAs) for HCV galvanised the World Health Organisation (WHO) to develop the first Global Health Sector Strategy (GHSS) to eliminate hepatitis C infection by 2030 [9]. Since then, many countries have been scaling up services to reach the WHO targets of diagnosing 90% of infections and treating 80% of diagnosed infections. Through doing this, it is hoped that they will achieve the ambitious elimination goals of reducing HCV incidence by 80% and reducing HCV-related mortality by 65% by 2030 compared to 2015. In light of this, Pakistan developed its strategic plan to eliminate HCV [10], including a provincial-level control program [11] and the Prime Minister's Programme to treat 9.8 million HCV patients by 2030 [8]. From 2005 to 2010, the National Hepatitis Control Program supported hepatitis C treatment and prevention activities in Pakistan [12], with 23,000 HCV patients treated [13]. After 2011, the provincial hepatitis and treatment programs coordinated HCV testing and treatment services [10]. DAA treatments were first available in 2014 and have been generally accessible since 2016 [10]. Despite this, the HCV treatment coverage in 2020 was estimated to be very low in the country, with 2% of infections having been treated [11].

The HCV cascade of care (CoC) is a framework that describes people's engagement in each step of the pathway of care from HCV diagnosis through linkage-to-care to successful treatment [14], and so allows monitoring of the progress made in scaling up testing and treatment. However, evidence on the HCV-CoC is sparse in low- and middle-income countries (LMICs), with a recent global systematic review finding only two good-quality estimates from LMICs (Egypt and Georgia) [15]. The Polaris group also estimated the HCV-CoC across different countries in 2020, using a range of methodologies depending on data availability, consisting of extrapolated estimates, drug sales data, data from surveillance and other databases [16]. However, data was sparse in LMICs, with estimates for levels of diagnosis being solely dependent on expert consensus or extrapolation from other countries for 40 of 62 reported LMICs and 20 of 55 reported countries for levels of HCV treatment. Additionally, there were only 4 of 22 countries from the Eastern Mediterranean Region that reported data on the HCV cascade of care in a 2019 review of the regional progress towards the WHO 2030 viral hepatitis elimination targets [17]. This emphasises the need for more good-quality estimates of the HCV-CoC in LMICs to determine gaps in the treatment pathway for guiding future initiatives and progressing

towards global HCV elimination. This is particularly important for countries with the highest burden of HCV infection, such as Pakistan, India, and China [18].

This study aimed to assess the HCV-CoC in Sindh Province, Pakistan's second-largest province and home to a 55.7 million population [19]. For individuals that have ever been eligible for HCV treatment (defined either as being currently infected (HCV-RNA positive) or previously been infected (HCV-Ab positive but HCV-RNA negative) and reporting being treated), we determined the proportion that have ever been tested for HCV, diagnosed, treated for HCV, and have evidence of being cured (undetectable HCV-RNA test result) using a serosurvey undertaken in 2019–2020. We also evaluated what factors are associated with ever being tested for HCV to determine which socio-demographic groups or groups with specific risk factors should be the target of future testing initiatives.

## Methods and materials

We utilised a household sero-survey undertaken in Sindh province from November 1, 2019, to June 15, 2020. The survey employed a two-stage stratified sampling technique. Initially, the province was stratified into urban and rural areas, with the urban areas further divided into large cities (Karachi, Hyderabad, and Sukkur) and other urban areas (all other cities and towns). The large cities were sub-stratified into low, middle, and high-income areas. The primary sampling units were enumeration areas (containing 200–250 households) in cities and towns or villages/mouzas/dehs in rural areas from each district, as prepared by the population census organisation. The survey sampled eight enumeration areas or villages/mouzas/dehs from each district, followed by five households from each sampling unit, with all available individuals from sampled households being included. The survey collected basic information from each individual, such as age, gender, family history of hepatitis (Hepatitis B or C), HBV vaccination status and and relationship with the head of the household (a person acknowledged as the reference point of the household (a group of related or unrelated persons who live together in the same dwelling unit) and shares the housekeeping arrangements [20]), with this data being collected from 6672 participants (2988 children (<18) and 3684 adults). Additional data on education, risk factors linked to HCV infection, previous HCV testing history, previous test results, and previous HCV treatment history were collected from 3684 adults (age ≥ 18 years) through an interview. No questions were asked on the type of test, whether the test result was confirmed, the type of treatment they received or whether they were treatment compliant. Written consent was taken from every sampled individual. Lastly, finger-prick blood samples were taken from all individuals for undertaking rapid HCV antibody tests using the SD-Bioline kit (details of sensitivity and specificity are given in previous studies [21,22]). Post-test counselling was provided for all individuals who tested HCV antibody (HCV-Ab) positive, and a venous blood sample was taken. This blood sample was stored in a gel tube, labelled and kept in a cool box after clotting. The gel tube was centrifuged to extract the serum for nucleic acid testing (NAT) using GeneXpert, with individuals testing HCV-RNA positive being linked to the hepatitis programme for treatment.

The survey team revisited each sampled household two times on consecutive days to include household members who were not present at the first visit. If an individual could not be included despite repeated visits, he/she was counted as absent (non-response). Children (defined as <18) were not included in this analysis because data on HCV risk factors and the HCV-CoC were not collected from them.

### Statistical analysis

STATA 18 was used to analyse the data. We firstly determined the proportion of all adults that self-reported previously being tested for HCV in the sero-survey dataset, and how long ago their last test occurred. We evaluated the level of concordance between self-reported test results and HCV-Ab test results from the sero-survey among those who reported their previous HCV test result. Binary logistic regression was then used to determine what factors are associated with self-reporting ever being tested for HCV to evaluate whether specific population sub-groups may have been missed by previous testing strategies. First, bivariable regression analysis was used to determine associations between ever being HCV

tested and either socio-demographic or various risk factors linked to HCV infection. Following this, multivariable logistic regression was used, only including those factors that had an association in the bivariable analysis (p-value<0.25) based on the p-value cut-off point suggested in the literature that discusses the purposeful variable selection technique [23]. We reported the unadjusted Odds Ratios (OR) and adjusted Odds Ratios (aOR) with 95% confidence intervals (95%CI) and p-value.

Secondly, we identified all adults who have ever been eligible for HCV treatment in the sero-survey dataset, defined as either being currently infected (test HCV-Ab positive and HCV-RNA positive) or having previously been infected (test HCV-Ab positive but HCV-RNA negative) and reporting being treated in the survey. We then determined the HCV-CoC among this group. We estimated the percentages of these treatment-eligible adults who reported previously being tested for HCV, reported testing positive for their previous HCV test (i.e., are diagnosed), reported receiving HCV treatment following their previous positive test result, and the percentage of treated individuals that had an undetectable HCV-RNA test result in the survey (i.e., evidence of successful treatment). People that tested HCV-Ab positive but did not have an HCV-RNA test were not included in this analysis (Fig 1).

Lastly, we undertook a sensitivity analysis in which we restricted our analysis to individuals who had attended at least primary education to examine how education may affect our HCV cascade of care results (S1 Text).

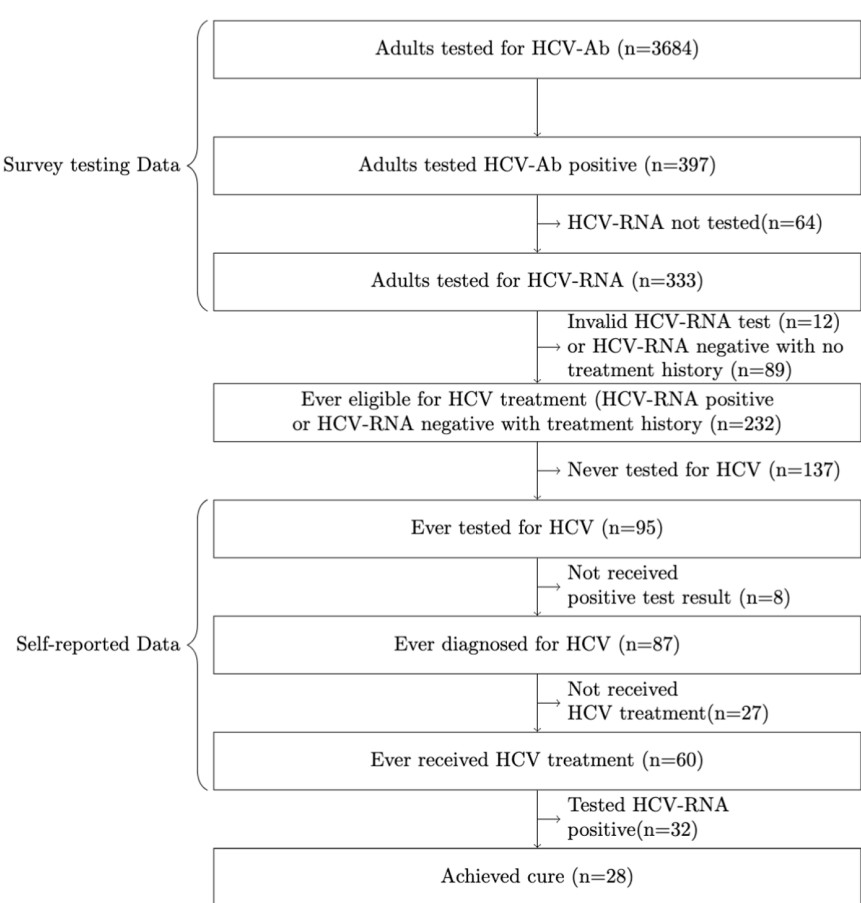

**Fig 1. Hepatitis C virus cascade of care framework.**

## Ethics approval

Ethical approval for the 2019–2020 Sindh Province Household Serosurvey was obtained from the National Bioethics Committee (NBC) Pakistan (Ref: No. 4–87/NBC-431/19/708). The data was collected from 01/11/ 2019 to 15/06/2020. Anonymised data from this Household Serosurvey was accessed on June 2, 2022.

## Results

Overall, 3684 adults had available HCV-Ab test results in the 2019–2020 household sero-survey. Of these, 808 (21.9%) were aged 18–24 years, 1997 (54.2%) were females and half (1845; 50.1%) had never been to school. In terms of risk factors linked to HCV infection, 2623 (71.5%) of adults reported therapeutic injection(s) in the last year, 793 (21.6%) reported being hospitalised in the last year, 290 (7.9%) had ever received a blood transfusion, 455 (12.4%) had ever received an invasive dental procedure, and 488 (13.3%) had ever received a tattoo or acupuncture (Table 1).

Nearly 10% of adults (341/3684 or 9.3%) in the survey reported previously being tested for HCV, of which 46.0% (157/341) had their last HCV test in the preceding one to five years and 39.3% (134/341) reported their previous test result was positive (Table 1).

**Table 1. Background characteristics of adults who participated in the 2019-2020 household sero-survey in Sindh province, Pakistan.**

| Variable | Category | n | Frequency | Percentage |
|---|---|---|---|---|
| Age (in years) | 18-24 | 3684 | 808 | 21.9 |
| | 25-64 | 3684 | 2616 | 71.0 |
| | ≥65 | 3684 | 260 | 7.1 |
| Gender | Female | 3684 | 1997 | 54.2 |
| | Male | 3684 | 1687 | 45.8 |
| Education status | Illiterate | 3684 | 1845 | 50.1 |
| | Primary | 3684 | 622 | 16.9 |
| | Secondary or higher | 3684 | 751 | 20.4 |
| | Unknown | 3684 | 466 | 12.7 |
| Family history of hepatitis | Yes | 3666 | 444 | 12.1 |
| Last year therapeutic injection history | Yes | 3669 | 2623 | 71.5 |
| Last year hospitalization history | Yes | 3665 | 793 | 21.6 |
| Ever shaved by traditional barber | Yes | 3521 | 991 | 28.2 |
| Ever blood transfusion history | Yes | 3651 | 290 | 7.9 |
| Ever-invasive dental procedure history | Yes | 3672 | 455 | 12.4 |
| Ever tattoo/acupuncture history | Yes | 2672 | 488 | 13.3 |
| Ever injectable drug use history | Yes | 3664 | 31 | 0.9 |
| Current health care worker | Yes | 3669 | 10 | 0.3 |
| Ever HBV vaccination | Yes | 3630 | 110 | 3.0 |
| Ever traditional Medicine (Hijama) | Yes | 3668 | 17 | 0.5 |
| Ever been tested for HCV | No | 3684 | 3290 | 89.3 |
| | yes | 3684 | 341 | 9.3 |
| | Unknown | 3684 | 36 | 1.0 |
| | Missing | 3684 | 17 | 0.5 |
| Timing of the last hepatitis C test (asked of those that report ever been tested for HCV) | last year | 341 | 81 | 23.8 |
| | 1-5 years | 341 | 157 | 46.0 |
| | >5 years | 341 | 95 | 27.9 |
| | Don't know | 341 | 8 | 2.4 |

## Factors associated with reporting ever being Hepatitis C tested

In the bi-variable unadjusted analyses, variables associated with ever HCV testing at p<0.25 included age, gender, educational status, history of being shaved by a traditional barber, ever had a tattoo/acupuncture, ever received a blood transfusion, ever received an invasive dental procedure, family history of hepatitis, ever received HBV vaccination, ever been hospitalised and receiving a therapeutic injection in the last year (Table 2). These were included in the multivariable model.

The multivariable model suggested that males (aOR=1.5, 95%CI:1.0-2.1), adults with secondary or higher educational level (aOR=2.3, 95%CI: 1.6-3.2), adults aged 25–64 years old (aOR=5.1, 95%CI: 3.1-8.6) or 65+years (aOR=2.6, 95%CI:1.2-5.5) were more likely to report ever being tested for HCV. Additionally, adults who had a family history of hepatitis (aOR=4.7, 95%CI:3.4-6.3), ever had tattoos/acupuncture (aOR=1.5; 95%CI:1.0-2.2), ever received blood transfusions (aOR=2.2, 95%CI:1.5-3.3), had been hospitalised in the last year (aOR=1.8, 95%CI:1.3-2.6), received therapeutic injections in the past year (aOR=1.5, 95%CI: 1.1-2.1), or ever received HBV vaccination (aOR=8.9, 95%CI:5.5-14.4) had a higher likelihood of reporting ever being HCV tested (Table 2).

## HCV cascade of care among treatment-eligible adults

Overall, 397 adults (397/3684 or 10.8%, 95%CI: 9.8-11.8%) tested HCV-Ab-positive in the 2019–2020 survey, with 321 (321/397 or 80.9%) having a valid HCV-RNA test result in the survey dataset. There was over 84% concordance between antibody test results and self-reported HCV status (Table A in S1 Text). Of 321 adults with HCV-RNA test results, 232 were defined as treatment eligible, with 204 (204/232 or 87.9%) still having HCV infection (HCV-RNA positive) and 28 (28/232 or 12.1%) no longer having HCV infection (HCV-RNA negative) and reporting previous treatment. The remaining 89 (89/321 or 27.7% of those tested for HCV-RNA) that tested HCV-Ab positive and were HCV-RNA negative reported no previous treatment, and so were assumed to have spontaneously cleared their infection (Table B in S1 Text).

Of the treatment-eligible adults, 95 (95/232 or 40.9%, 95%CI: 34.6-47.6%) reported ever being tested for HCV, with 48 (48/95 or 50.5%) saying they were last tested 1–5 years ago and 16 (16/95 or 16.8%) tested in the last year. Of tested individuals, 87 (87/95 or 91.2%, 95% CI: 84.1- 96.3%) reported their last HCV test was positive, and 60 (60/87 or 69%, 95%CI: 58.1%-78.5%) reported receiving HCV treatment. This equates to 37.5% (87/232: 95%CI: 31.3-44.1%) of treatment eligible adults reporting that they have ever been diagnosed for HCV and 25.9% (60/232; 95%CI: 20.4-32.0%) of treatment eligible adults reporting they have ever received HCV treatment, with 28 (28/60 or 46.7%, 95%CI: 33.7-60.0%) of treated individuals having undetectable HCV-RNA levels (Fig 2).

## Discussion

The 2019–2020 Sindh household sero-survey estimated that 10.8% of adults from Sindh province have been exposed to HCV (tested HCV-Ab positive in the 2019–2020 survey). Although the overall HCV testing rate in the province was low, with only 9.3% of adults reporting previously being tested for HCV, testing rates were much better among adults that are or have ever been eligible for treatment, with 40.9% reporting previously being tested, of which 91.2% reported ever being diagnosed HCV positive and 69.0% of diagnosed individuals reporting ever having HCV treatment. This translates to 37.5% of treatment-eligible adults reporting ever being diagnosed with HCV, 25.9% of treatment-eligible adults reporting ever receiving HCV treatment, and 46.7% of these adults then testing RNA negative in the survey, suggestive of an SVR result. HCV testing rates were also higher among adults that have ever been exposed to HCV (29.2%) or are currently infected (32.8%) and were also higher among males, older adults, adults with better education, who had received HBV vaccination, and adults with risk factors linked to HCV infection. This included having a family history of hepatitis, ever having a blood transfusion or tattoo/acupuncture, a history of hospitalisation, or therapeutic injections in the last year.

**Table 2. Factors associated with self-reporting ever being tested for Hepatitis C among adults in the 2019-2020 survey.**

| Variable | Odds ratio (OR) | p-value | Adjusted odds ratio (aOR) | p-value |
|---|---|---|---|---|
| **Age group (in years)** | | | | |
| 18-24 | 1 | | 1 | |
| 25-64 | 4.9 (3.1-7.7) | <0.001 | 5.1 (3.1-8.6) | <0.001 |
| ≥65 | 2.8 (1.5-5.4) | 0.002 | 2.6 (1.2-5.5) | 0.014 |
| **Gender** | | | | |
| Female | 1 | | 1 | |
| Male | 1.2 (1.0-1.5) | 0.078 | 1.5 (1.0-2.1) | 0.043 |
| **Educational status** | | | | |
| Illiterate | 1 | | 1 | |
| Primary | 1.5 (1.1-2.1) | 0.008 | 1.4 (0.9-1.8) | 0.214 |
| Secondary or higher | 2.2 (1.7-2.9) | <0.001 | 2.3 (1.6-3.2) | <0.001 |
| Unknown | 0.6 (0.4-1.0) | 0.047 | 1.1 (0.7-1.9) | 0.611 |
| **Ever shaved by traditional barber** | | | | |
| No | 1 | | 1 | |
| Yes | 1.4 (1.1-1.8) | 0.003 | 0.9 (0.6-1.3) | 0.553 |
| **Ever tattoo/acupuncture history** | | | | |
| No | 1 | | 1 | |
| Yes | 1.2 (0.9-1.7) | 0.223 | 1.5 (1.0-2.2) | 0.035 |
| **Family history of hepatitis** | | | | |
| No | 1 | | 1 | |
| Yes | 4.9 (3.8-6.3) | <0.001 | 4.7 (3.4-6.3) | <0.001 |
| **Last year hospitalization history** | | | | |
| No | 1 | | 1 | |
| Yes | 3.0 (2.4-3.8) | <0.001 | 1.8 (1.3-2.6) | <0.001 |
| **Ever blood transfusion history** | | | | |
| No | 1 | | 1 | |
| Yes | 3.8 (2.8-5.1) | <0.001 | 2.2 (1.5-3.3) | <0.001 |
| **Ever invasive dental procedure history** | | | | |
| No | 1 | | 1 | |
| Yes | 2.6 (1.9-3.4) | <0.001 | 1.3 (0.9-1.8) | 0.155 |
| **Last year therapeutic injection history** | | | | |
| No | 1 | | 1 | |
| Yes | 1.4 (1.1-1.9) | 0.007 | 1.5 (1.1-2.1) | 0.014 |
| **Ever injection drug use history** | | | | |
| No | 1 | | | |
| Yes | 1.1 (0.3-3.4) | 0.960 | | |
| **Ever traditional medicine(Hijama)** | | | | |
| No | 1 | | 1 | |
| Yes | 3.0 (1.0-9.3) | 0.055 | 0.5 (0.1-2.7) | 0.407 |
| **Ever received HBV vaccine** | | | | |
| No | 1 | | 1 | |
| Yes | 12.1 (8.1-17.9) | <0.001 | 8.9 (5.5-14.4) | <0.001 |

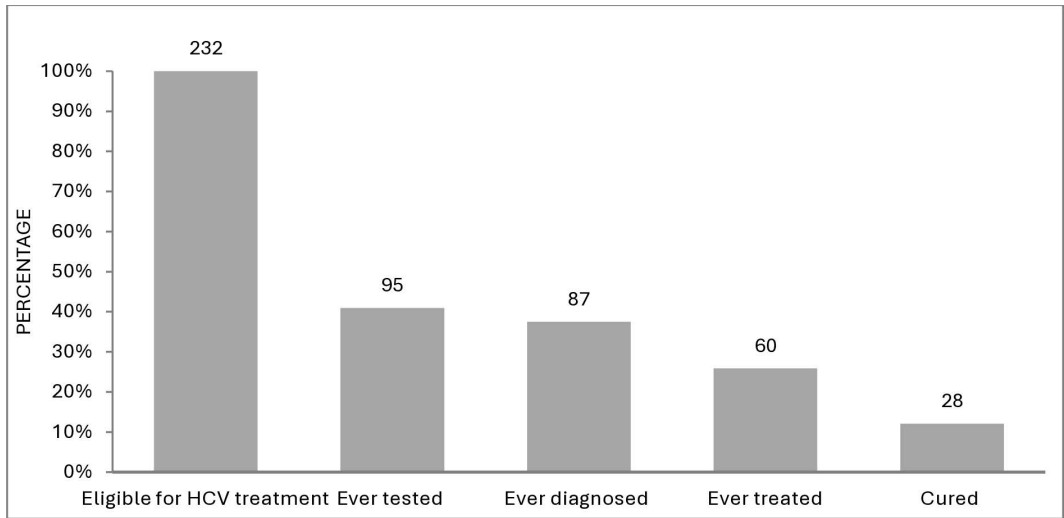

**Fig 2. Hepatitis C Cascade of Care among adults that have ever been eligible for HCV treatment in the 2019-2020 household sero-survey in Sindh province, Pakistan.** We define adults that have ever been eligible for HCV treatment as either being currently infected (test HCV-Ab positive and HCV-RNA positive) or having previously been infected (test HCV-Ab positive but HCV-RNA negative) and self-reporting ever being treated in the survey.

## Strengths and limitations

This is the first study that we know to evaluate the cascade of care for HCV in Pakistan and the factors associated with HCV testing among adults (≥18 years). The study utilised a large community-based household survey that took a representative sample from urban and rural areas in Sindh Province, enhancing the generalisability of our findings.

Limitations mainly involve the reliance on self-reported data for determining history of HCV testing, diagnosis and treatment history. Although our findings are broadly similar when we consider literate participants (the HCV testing coverage was slightly higher as expected, 13.1% among individuals who attended at least primary education versus 9.3% overall. Regarding the HCV cascade of care, from 62 treatment-eligible individuals who attended at least primary education, 53.2% reported ever being tested vesrsus 41.2% overall, 93.9% reported testing positive versus 91.6% overall, and 74.2% reported receiving HCV treatment versus 69.0% overall), this might affect the accuracy of our estimates, particularly among those who were illiterate (50% of the sample). Importantly, of those that tested HCV-Ab negative, we found that 15.5% reported a previous positive HCV diagnosis, while 13.0% of those testing HCV-Ab positive reported a previous negative diagnosis. Some level of discordance should be expected due to some participants not fully understanding what HCV is and/or getting confused between testing for HCV and testing for other forms of hepatitis (e.g., hepatitis B) or other infections, especially if the test had been done years ago. This concern is supported by there being higher levels of discordant results among survey participants that were illiterate compared to those that have secondary or higher education (19.9% versus 8.1% overall discordance levels), highlighting the importance of improving health education in this group. We also have no information on the accuracy of the tests used among individuals self-reporting previously being HCV tested. The tests may have had poor sensitivity and specificity compared to the SD-Bioline test used in this survey [21,22,24], or the SD-Bioline test may have had some issues of accuracy in some survey settings, possibly due to such things as low temperature [22,25]. It is also possible that some of the observed discrepancies in individuals testing HCV-Ab negative are due to individuals losing their antibody response after undergoing treatment, as observed in Australia following the scale-up in DAA treatment [26]. In support of this, 22 of the 33 individuals with discordant results among adults testing Ab-negative in our survey reported ever having HCV treatment, and so sero-reversion could explain some of the discrepancies. Other studies of the general population [27–29] and individuals infected with HIV [30–32] have also

observed this, with levels of discordance varying from 6.3% to 25.8%. It has been hypothesised that full or partial sero-reversion could occur due to loss of antigenic stimulation or low viraemia levels that are insufficient to maintain humoral immune responses [28,33], especially among immunocompromised [34] and treated individuals [31,32,35].

Conversely, for discrepancies among individuals testing Ab-positive, it is possible that the discrepancy could be due to individuals becoming infected since their last reported negative test. Reinfection could also partially explain the low proportion of testing RNA-negative (46.7%) among individuals who reported ever having treatment, although it could also be due to the low SVR from using old interferon-based treatments [12,36]. However, we were unable to confirm this due to a lack of data on treatment compliance or detailed treatment regimens among individuals who self-reported previous HCV treatment.

Lastly, the sampling unit for this survey was households. Because of this, the survey may not be representative of high-risk populations because it will under-sample high-risk populations such as homeless people, people who inject drugs (PWIDs), migrants, and prisoners [37].

## Comparison with previous studies

The overall proportion of adults reporting ever being tested for HCV (9.3%) in this study is consistent with what was reported in a household serosurvey undertaken in Punjab in 2018 [38], which estimated that 8.0% of adults in Punjab had ever been diagnosed with Hepatitis B or C. In contrast, the 2017/18 Demographic Health Survey (DHS) for Pakistan suggested a higher rate of ever testing (25.4%) for HBV or HCV among reproductive-aged women [39]. Unfortunately, neither survey distinguished between Hepatitis B and C. The higher rate of testing in the Pakistan DHS may be due to it being undertaken among reproductive-aged women who are likely to have a higher HBV testing rate due to the Pakistan policy of HBV screening among pregnant women [10]. Otherwise, our levels of diagnosis agree with other estimates. We found that 37.5% of adults that have ever been eligible for treatment reported ever being diagnosed with HCV, while 28.9% of adults with chronic HCV infection (test RNA-positive) reported ever being diagnosed in 2019–2020. This is similar to a modelled estimate that 22% of persons living with HCV in Pakistan were diagnosed in 2020 [8].

## Conclusion and implications

Our analysis suggests moderate levels of HCV diagnosis and treatment in Sindh province in 2019–2020. Levels of HCV diagnosis and treatment are likely to have improved since then considering the ongoing HCV testing and treatment initiatives being undertaken in Sindh [6], although COVID-19 posed significant challenges to viral hepatitis services in Pakistan [40,41]. Given the increasing prevalence of HCV exposure among adults in the Province [6], there is a need to maintain and scale-up existing HCV diagnosis and treatment services to reduce the expanding HCV epidemic and achieve the WHO targets of diagnosing 80% of HCV infections and treating 65% of HCV infections in Sindh province and Pakistan. Adults with risk factors linked to being HCV infected (e.g., having a family history of hepatitis, blood transfusion, tattoo/acupuncture, hospitalisation, or therapeutic injection history) were more likely to be tested for HCV, which suggests existing HCV testing strategies target individuals with higher levels of risk, as confirmed by the high level of diagnosis among those with chronic HCV infection. The HCV testing rate was higher among individuals who had received HBV vaccination. The provincial HCV guidelines for Punjab, Pakistan, recommend HBV screening and HBV vaccination (if negative) for individuals with HCV infection [42]. This recommendation is also practised in Sindh and is likely to explain the strong associations of HBV vaccination with HCV testing. Education is strongly linked to healthcare access and health literacy in Pakistan, with individuals with higher educational levels having better access to healthcare and improved health literacy [43], thereby facilitating HCV testing. The higher HCV testing coverage among older adults could be explained by previous risk-based screening, given the higher prevalence of HCV infection in this group, as well as the effect of age, which will increase the chance of HCV testing as people grow older.

Women, young adults and individuals with low levels of education had lower levels of HCV testing highlighting that future initiatives need to target these groups, especially illiterate individuals seeing they also have a higher prevalence of HCV infection [6]. Studies suggest that financial incentives could promote HCV testing, so this could be considered in these groups [44]. HCV testing should be improved among women to prevent HCV mother-to-child transmission, which is an important route of HCV transmission for children in Pakistan [45]. The lower HCV testing coverage among women could occur for two reasons. Firstly, women in Pakistan have restricted freedom, and so many women remain at home [46,47]. This could restrict them being tested in community settings, while being tested through outreach activities may also be restricted because it is culturally inappropriate for male health workers to enter the home [48], and so they would not be tested if a female health worker is not available. Secondly, they may refuse testing due to fear of stigma and possible abandonment or divorce if they were found to be infected [49,50]. Antenatal HCV screening could be a good strategy for enhancing HCV testing among women in Pakistan [51], which has high acceptability [52]. Through doing these strategies, Pakistan should be able to achieve HCV elimination as recently achieved by Egypt [53].

## Supporting information

**S1 Text. This file contains sensitivity analysis results and two supplementary tables (Table A and Table B).**
(DOCX)

**S1 Data. This file contains the dataset used to generate the results.**
(CSV)

## Author contributions

**Conceptualization:** Tesfa Sewunet Alamneh, Josephine G Walker, Aaron G Lim, Peter Vickerman.

**Data curation:** Ejaz Alam, Saeed Hamid, Graham R Foster, M. Azim Ansari, Huma Qureshi.

**Formal analysis:** Tesfa Sewunet Alamneh.

**Funding acquisition:** Josephine G Walker, Aaron G Lim, Peter Vickerman.

**Software:** Tesfa Sewunet Alamneh.

**Supervision:** Josephine G Walker, Aaron G Lim, Peter Vickerman.

**Writing – original draft:** Tesfa Sewunet Alamneh.

**Writing – review & editing:** Tesfa Sewunet Alamneh, Josephine G Walker, Aaron G Lim, Ejaz Alam, Saeed Hamid, Graham R Foster, Naheed Choudhry, M. Azim Ansari, Huma Qureshi, Peter Vickerman.

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
