## [Decision Letter · Decision Letter 0]

PGPH-D-25-00297

Hepatitis C virus Cascade of Care among adults in Sindh province, Pakistan: Findings from 2019 household sero-survey

Dear Dr. Alamneh,

Thank you for submitting your manuscript to PLOS Global Public Health. After careful consideration, we feel that it has merit but does not fully meet PLOS Global Public Health’s publication criteria as it currently stands. Therefore, we invite you to submit a revised version of the manuscript that addresses the points raised during the review process.

We look forward to receiving your revised manuscript.

Kind regards,

Kévin Jean

Academic Editor

Additional Editor Comments (if provided):

Dear Author,

Many thanks for your submission, which has been assessed by 2 independent reviewers.

As you will see, both reviewers highlighted some methodological issues and perhaps a need to further develop some parts of the context and of the discussion. This motivated the decision for Major revision.

As you will see, R2 also raises some issues regarding the perceived novelty of your study. As PLOS GPH's position is to considering only methodological rigor and ethical standards, the consideration of perceived novelty are not to be regarded. Therefore, this criteria will not be considered for the final decision, although this comment also echoes the fact that your paper may benefit for more developped discussion and contextualisation.

All the best,

Reviewers' comments:

Reviewer's Responses to Questions

**Comments to the Author**

1. Does this manuscript meet PLOS Global Public Health’s publication criteria?

Reviewer #1: Yes

Reviewer #2: No

2. Has the statistical analysis been performed appropriately and rigorously?

Reviewer #1: I don't know

Reviewer #2: Yes

3. Have the authors made all data underlying the findings in their manuscript fully available (please refer to the Data Availability Statement at the start of the manuscript PDF file)?

Reviewer #1: Yes

Reviewer #2: Yes

4. Is the manuscript presented in an intelligible fashion and written in standard English?

Reviewer #1: Yes

Reviewer #2: No

Reviewer #1: General comment

This study presents results from a sero-survey conducted in Pakistan (Sindth Province) and aims at i. assessing the Cascade of Care linked to HCV among 3684 adults and ii. evaluating the factors linked to self-reporting of HCV testing. It shows a low testing rate among this sample of less than 10%, provides an estimate of the CoC in this province and highlights factors associated with reporting HCV testing, that could guide future testing strategies in this region. However, some issues need to be addressed before considering publication of this manuscript. A list of my further comments is provided below.

Introduction

L51 : One could add additional references for the risk of transmission through healthcare practices (see https://onlinelibrary.wiley.com/doi/full/10.1111/apt.17106, a recent meta-analysis of HCV virus infection associated with different hospital-based procedures) , and IDU (https://www.thelancet.com/journals/langas/article/PIIS2468-1253(19)30085-8/abstract for example)

L80: Please include references focusing on the prevalence of HCV in these countries.

Material and methods

L90 : It is stated that the sero-survey was carried-out in 2019 but, as mentioned in the ethical approval, “data was collected from 01/11/2019 to 15/06/2020”.

L99 : Can you explain the meaning of “head of the household” ?

L106 : Can you provide sensitivity and specificity values associated with this test kit, or provide a reference associated with this kit that mentions them?

L108-109 : Consider rephrasing this part of the sentence.

Statistical analysis

What kind of tool did you use to conduct these analyses ? Please give information on the software used. You might also consider providing the associated code.

L116 : I think that this information is deducted directly from empirical data, “estimated” might not be the right term.

L124 : How did you decide on the 0.2 threshold?

Results

Can you report which variables were retained for the multivariate analysis based on unadjusted ORs? - Consider either reporting unadjusted ORs in the text of the Results section or keeping only aORs and adding unadjusted ORs as Supplementary Information.

The p-value associated with the unadjusted OR of tattoo and acupuncture is of 0.223 but this variable was retained in the multivariate model.

HBV vaccination is mentioned as a risk factor but does not appear in the M&M section.

Discussion

Can you comment on the history of HBV vaccination, which appears as the variable with the highest associated aOR .

The comparison with other studies focuses on the survey itself but not on the highlighted risk factors; did you expect such results? what would be the intuition behind these results?

L256 : “Adults with HCV risk factors were more likely to be tested for HCV”. Maybe clarify that “HCV risk factors” refers to the risk factors linked with HCV infection/positivity. This part seems a bit vague, can you provide a list of some of these factors ?

Some sentences may need to be rephrased for clarity.

Reviewer #2: Thank you for the opportunity to review this manuscript “Hepatitis C virus Cascade of Care among adults in Sindh province, Pakistan: Findings from 2019 household sero-survey”.

The authors conducted a household serosurvey to describe the cascade of care for HCV among adults in Sindh Province, Pakistan. While the topic of HCV CoC is indeed relevant to global public health, particularly in regions with high endemicity such as Pakistan, several significant issues limit the originality and contribution of this manuscript to the existing literature:

o The cascade-of-care model for HCV is well-established, and similar studies have been extensively conducted globally, including in comparable LMIC contexts. This manuscript presents findings that largely align with previous reports from Pakistan and other regions, thus offering limited novel insights.

o Although the authors identified demographic factors (e.g., age, gender, education) associated with gaps in testing and treatment, the manuscript does not offer an in-depth exploration or innovative interpretation of these disparities. The explanations provided are cursory and do not sufficiently advance understanding beyond established knowledge.

o Reliance on self-reported data introduces significant potential bias, particularly in a population with high illiteracy rates. This limitation compromises the reliability of the cascade-of-care assessment. Additionally, no information on treatment compliance or detailed treatment regimens limits the utility of findings for practical interventions.

o While the discussion broadly notes the need for improvement, it fails to provide specific, actionable, and innovative recommendations that could guide public health interventions or policy adjustments.

o Minor comments: Redundant repetition of results (particularly evident in the abstract) detracts from clarity and readability, and the manuscript would benefit from substantial editorial refinement.

**Do you want your identity to be public for this peer review?** For information about this choice, including consent withdrawal, please see our Privacy Policy

Reviewer #1: **Yes: ** Paul Henriot

Reviewer #2: No

---

## [Decision Letter · Decision Letter 1]

Hepatitis C virus Cascade of Care among adults in Sindh province, Pakistan: Findings from 2019-2020 household sero-survey

PGPH-D-25-00297R1

Dear Mr Alamneh,

We are pleased to inform you that your manuscript 'Hepatitis C virus Cascade of Care among adults in Sindh province, Pakistan: Findings from 2019-2020 household sero-survey' has been provisionally accepted for publication in PLOS Global Public Health.

Best regards,

Kévin Jean

Academic Editor

Dear Authors,

Many thanks for your efforts and edits, which were appreciated by both Reviewers. We are happy to accept your draft for publication.

All the very best,

Reviewer Comments (if any, and for reference):

Reviewer's Responses to Questions

**Comments to the Author**

Reviewer #1: All comments have been addressed

Reviewer #2: All comments have been addressed

publication criteria?

Reviewer #1: Yes

Reviewer #2: Yes

3. Has the statistical analysis been performed appropriately and rigorously?

Reviewer #1: Yes

Reviewer #2: Yes

4. Have the authors made all data underlying the findings in their manuscript fully available (please refer to the Data Availability Statement at the start of the manuscript PDF file)?

Reviewer #1: Yes

Reviewer #2: Yes

5. Is the manuscript presented in an intelligible fashion and written in standard English?

Reviewer #1: Yes

Reviewer #2: Yes

Reviewer #1: I thank the authors for these detailed responses. It seems that all of the issues raised have been addressed.

Reviewer #2: Thanks to the authors for performing the required modifications , which made the article better.

**Do you want your identity to be public for this peer review?** For information about this choice, including consent withdrawal, please see our Privacy Policy

Reviewer #1: No

Reviewer #2: No
